# Convergent adaptation of *Saccharomyces uvarum* to sulfite, an antimicrobial preservative widely used in human-driven fermentations

Laura G. Macías[1,2], Melisa González Flores[3,4], Ana Cristina Adam[1], María E. Rodríguez[3,5], Amparo Querol[1], Eladio Barrio[1,2], Christian Ariel Lopes[3,4], Roberto Pérez-Torrado[1]*

1 Instituto de Agroquímica y Tecnología de los Alimentos, IATA-CSIC, Paterna, Spain, 2 Departament de Genètica, Universitat de València, Valencia, Spain, 3 Instituto de Investigación y Desarrollo en Ingeniería de Procesos, Biotecnología y Energías Alternativas (PROBIEN, Consejo Nacional de Investigaciones Científicas y Técnicas de la República Argentina–Universidad Nacional del Comahue), Neuquén, Argentina, 4 Facultad de Ciencias Agrarias, Universidad Nacional del Comahue, Cinco Saltos, Río Negro, Argentina, 5 Facultad de Ciencias Médicas, Universidad Nacional del Comahue, Cipolletti, Río Negro, Argentina

* rober@iata.csic.es

**Data Availability Statement:** Raw sequencing data of S. uvarum strains sequenced in this study were submitted under BioProject PRJNA471597.

## Abstract

Different species can find convergent solutions to adapt their genome to the same evolutionary constraints, although functional convergence promoted by chromosomal rearrangements in different species has not previously been found. In this work, we discovered that two domesticated yeast species, *Saccharomyces cerevisiae*, and *Saccharomyces uvarum*, acquired chromosomal rearrangements to convergently adapt to the presence of sulfite in fermentation environments. We found two new heterologous chromosomal translocations in fermentative strains of *S. uvarum* at the *SSU1* locus, involved in sulfite resistance, an antimicrobial additive widely used in food production. These are convergent events that share similarities with other *SSU1* locus chromosomal translocations previously described in domesticated *S. cerevisiae* strains. In *S. uvarum*, the newly described VII^XVI and XI^XVI chromosomal translocations generate an overexpression of the *SSU1* gene and confer increased sulfite resistance. This study highlights the relevance of chromosomal rearrangements to promote the adaptation of yeast to anthropic environments.

## Author summary

It is known that genetically distant species can arrive to similar evolutionary solutions during the adaptation to a specific environment, a phenomena known as convergent adaptation, and this frequently occurs after point mutations, gene duplications, or species hybridizations. In this work, we discovered a new example of convergent evolution in the adaptation of two wine fermentation yeast species to the presence of sulfite, an antimicrobial additive widely used in food production. We observed that two species,

**Funding:** This work was supported by grants from the Ministerio de Ciencia, Innovación y Universidades to A.Q (RTI2018-093744-B-C31), EB (RTI2018-093744-B-C32), and by Conselleria d'Educació, Investigació, Cultura i Esport grant PROMETEO/2020/014 to A.Q and EB, as well as grants PICT 2015-1198 from the Fondo para la investigación Científica y Tecnológica, PIP 2015-555 from Comisión de Investigaciónes Científicas and PI04-A128 from Universidad Nacional de Comahue to CL. MGF also thanks CONICET for a postdoctoral fellowship. The funders had no role in study design, data collection and analysis, decision to publish, or preparation of the manuscript.

**Competing interests:** The authors have declared that no competing interests exist.

*Saccharomyces cerevisiae* and *Saccharomyces uvarum*, acquired chromosomal rearrangements to convergently adapt to the presence of sulfite in fermentative environments. We describe new chromosomal translocations in *S. uvarum* strains that generate an overexpression of the *SSU1* gene and confer increased sulfite resistance, a similar event that was already described in *S. cerevisiae*. Overall, this study describes a new case of convergent evolution in which the chromosomal rearrangements have a significant role in the adaptation of yeast to an environment created by humans to produce food.

## Introduction

Organisms belonging to different lineages can evolve independently to overcome similar environmental pressures through different molecular mechanisms. This convergent evolution has been seen as evidence of the action of natural selection [1,2]. In recent years, comparative genomics studies have suggested that convergent adaptations occur more frequently than previously expected [3,4]. For example, species of insects spanning multiple orders have independently evolved higher tolerance to toxic compounds produced by plants after different amino acid substitutions that might lower sensitivity to cardenolides [5] demonstrating that convergent adaptation can occur in nature between organisms belonging to different taxonomic levels. In the case of yeasts, convergent evolution by point mutations has been described both in evolving yeast species in nature [6] and in short-term evolutionary studies in the species *Saccharomyces cerevisiae* [7], for example in populations evolved under glucose limitation that increased fitness after alternative mutations in the genes *MTH1* and *HXT6/HXT7* [8]. Convergent evolution can occur through different mechanisms, including point mutations, gene duplications, and species hybridizations. Examples of convergent evolution via chromosomal rearrangements are rare, a single study has suggested that an intrachromosomal translocation is responsible of a convergent evolution in independent lineages in the case of the major histocompatibility complex [9]. A second study has suggested that amylase evolution in fish may have converged though a putative chromosomal translocation, although this has not yet been confirmed [10].

The genus *Saccharomyces* is composed of eight species including the model organism *S. cerevisiae* [11]. There is a substantial nucleotide divergence displayed for example between *S. cerevisiae* and the species *S. uvarum* and *S. eubayanus*, comparable to the divergence found between humans and birds [12]. *S. cerevisiae* has traditionally been associated with food and beverage fermentations which have been traced back to 5,000–10,000 years ago [13,14]. This domestication of *S. cerevisiae* by humans has left footprints that characterize their genome [15,16,17]. Along with *S. cerevisiae*, the species *S. uvarum* is the only natural species of the *Saccharomyces* genus that shows ecological success in human-driven fermentative environments [18]. *S. uvarum* coexists and even replaces *S. cerevisiae* in wine and cider fermentations performed at low temperatures, in particular at regions with oceanic or continental climate [19–21]. Genomic footprints of domestication, like introgressions, have also been reported in *S. uvarum* genomes [22].

During fermentation processes, yeast cells face adverse conditions such as osmotic stress due to high sugar concentrations, low temperatures, low pH, and the presence of certain toxic compounds used as preservatives. One of the most common preservatives used in wine and cider fermentations is sulfite [23]. The most common molecular mechanism to deal with the presence of sulfite in the media in yeasts involves the sulfite efflux with a plasma membrane pump encoded by the gene *SSU1* [24,25]. The strains lacking this gene showed a higher

sensitivity to sulfite due to the intracellular accumulation of this compound [26]. The transcription factor encoded by the *FZF1* gene has been reported to interact with the upstream promoter region of the gene *SSU1* to increase its transcription [26].

Mutations causing large-scale chromosomal rearrangements often occur in yeast populations rather than less frequent small-scale changes [27]. Even though most large-scale changes are deleterious and, therefore, quickly removed from the population, these mutations contribute to the genetic variation within the population facilitating the rapid adaptation to novel environments [28,29]. It has been reported that specific chromosomal rearrangements in *S. cerevisiae* wine strains generate an overexpression of the *SSU1* gene that increases the tolerance to sulfite [30], although it has been suggested that other unrelated sulfite tolerance adaptations could be present in the genome of the wine strains [31]. A reciprocal translocation between chromosomes VIII and XVI replaced the promoter of the *SSU1* gene, encoding a sulfite transporter [30]. This modification causes an increased expression of *SSU1* and, as a consequence, a greater resistance to sulfite [30]. After this first evidence, several groups have confirmed both the presence of this rearrangement in different strains belonging to the *S. cerevisiae* wine yeast subpopulation and the advantage that sulfite resistance confers to yeasts during their competition in wine fermentation [32–34]. Translocation VIII[XVI] has been proposed not only to contribute to the ecological differentiation of wine yeasts but also to the partial reproductive isolation between wine and wild subpopulations of *S. cerevisiae* [35,36]. Years later, another translocation event, between chromosomes XV and XVI, was described and associated with an increase in the expression of the *SSU1* gene in *S. cerevisiae* [37]. Another molecular mechanism causing the overexpression of this gene found in *S. cerevisiae* is an inversion in chromosome XVI [38]. A recent study with hundreds of strains confirmed the dominant presence of these *SSU1* locus rearrangement in the wine strains population, specially in commercial starters [39].

The promoter region of the *SSU1* gene has been demonstrated to be a hotspot of evolution in *S. cerevisiae* leading to different chromosomal rearrangements with a common phenotypic outcome: an increased sulfite tolerance. This work aims to test the evidence of convergent evolution at a higher taxonomic level by using another *Saccharomyces* species isolated from human-driven environments, *S. uvarum*. In this study, several strains of *S. uvarum* isolated from a wide range of environments and geographic locations have been used to identify high sulfite tolerant strains and the underlying molecular mechanisms associated with this trait.

## Results

### Two new chromosomal translocation events in the *SSU1* promoter of *S. uvarum* strains

A total number of 21 *S. uvarum* genomes (S1 Table) were assembled and examined to find structural variations in the promoter of the *SSU1* gene. Assemblies allowed us to identify two candidate chromosomal rearrangements in the promoter of this gene located at chromosome XVI (Fig 1A). Annotated and assembled strains were evaluated for synteny conservation and manual comparison of the annotation of *SSU1* gene confirmed different chromosomal locations in different strains. One of them was found in the genomes of three fermentative strains (BMV58, CECT12600, and NPCC1417) and involves chromosome VII. The other rearrangement involves chromosome XI and it was found in the strain BR6-2 isolated from a fermentative environment [22]. Strains CECT12600 and BMV58 were isolated in Spain from wine fermentations, while BR6-2 and NPCC1417 were isolated from cider fermentations in France and Argentina respectively. These chromosomal rearrangements changed the genomic context in the upstream region of the *SSU1* gene (Fig 1B). Instead of the *NOG1* gene present in the

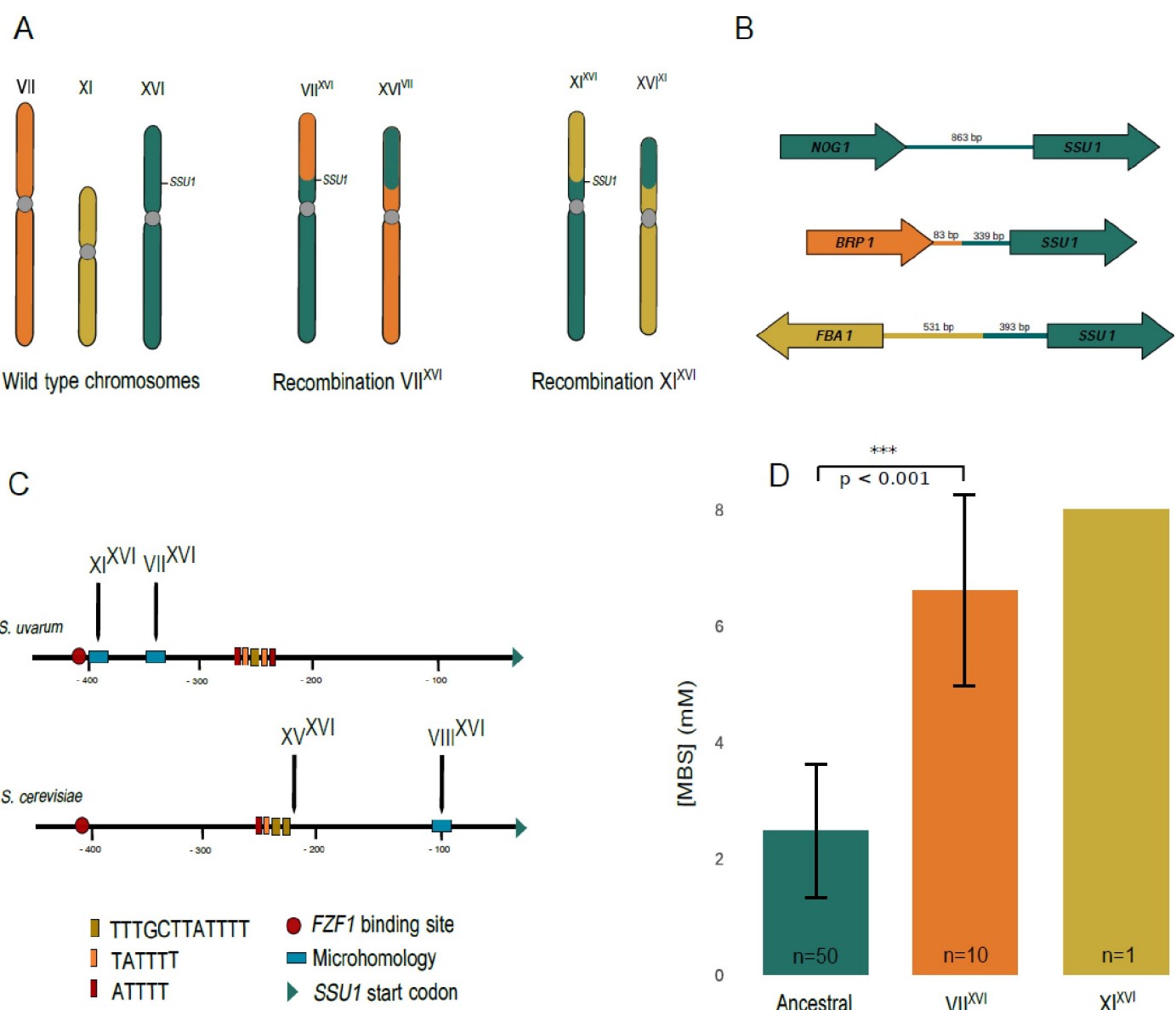

**Fig 1. New *SSU1* promoter variants found in *S. uvarum*.** Panel A. Ancestral type chromosomes; chromosomes VII and XVI after the reciprocal translocation in the *SSU1* promoter; chromosomes XI and XVI after the reciprocal translocation in the *SSU1* promoter. Panel B. Gene context surrounding the *SSU1* gene in the genomes with the ancestral and rearranged chromosomes. The distance between the *SSU1* gene and the previous gene is depicted in base pairs, in both the ancestral and recombinant genomes. Panel C. *SSU1* promoter and chromosomal translocation sites described for *S. uvarum*, in this study, and *S. cerevisiae* in previous [30,37]. *FZF1* binding site and microhomology sites are shown as well as the sites where the chromosomal translocation events occurred in both species reported. Panel D. Bar chart showing the tolerance to sulfites of the collection of *S. uvarum* strains tested by drop test assay. Ancestral strains: 52 strains without any of the two rearrangements reported; VII[XVI]: 10 strains with the chomosome VII and XVI rerrangement; XI[XVI]: one strain with the chromosome XI and XVI rearrangement. Tolerance to sulfite is measured by the maximum concentration of MBS in which cells can grow. The bars represent the mean of the maximum MBS concentration reached by each strain and the arrows represent the standard deviation. A *t*-test was performed between the strains having the ancestral *SSU1* promoter and the strains carrying the VII[XVI] rearrangement. We obtained a significant *p*-value < 0.001.

ancestral *SSU1* promoter strains, the recombinant chromosome VIII[XVI] has the *BRP1* gene and the XI[XVI] has the *FBA1* (gene reverse strand) upstream of *SSU1*. The rearrangement observed between chromosomes VII and XVI was identified at 339 bp upstream of the *SSU1* gene start (Fig 1B) within a microhomology region (Fig 1C) similarly to the VIII[XVI] transloca-tion described in *S. cerevisiae* strains. The distance between the end of this gene and the

beginning of the *SSU1* gene is 422 bp and 924 bp between the starts of both genes (Fig 1B). In the assembled genome of the *S. uvarum* BR6-2 strain, the rearrangement between chromosomes XI and XVI occurred at 393 bp upstream of the *SSU1* gene start also within a microhomology region (Fig 1C). Both *SSU1*-promoter chromosomal translocation events described in this study occurred before the *FZF1* binding site (Fig 1C), a well-known *SSU1* gene transcriptional regulator, indicating that this site has been lost in these strains, as also occurred in the two chromosomal translocation events described in *S. cerevisiae*.

To determine the frequency of these translocations in *S. uvarum*, we designed specific PCR tests to evaluate a collection of 64 *S. uvarum* strains obtained from different geographic locations and sources, including both natural and anthropic environments, such as wine and cider fermentations (S2 Table). The PCR amplification allowed us to identify if any of these strains carried any of the two rearrangements identified at the *SSU1* promoter. Rearrangements between chromosomes VII and XVI were found in a total number of 10 strains while the rearrangement involving chromosomes XI and XVI, was only identified in the BR6-2 strain (S4 Table and S4–S9 Figs). Southern blot method was used to classify the most frequent chromosomal rearrangement (VII$^{XVI}$) as a reciprocal chromosomal translocation (S1 Fig). Finally, PacBio end-to-end genome assembly of BR6-2 revealed that the rearrangement between the chromosomes XI and XVI also corresponds to a reciprocal translocation (Fig 1A).

## Strains carrying the chromosomal rearrangements in the *SSU1* promoter are more tolerant to sulfite

Sulfite tolerance was evaluated by drop test assays in the 64 *S. uvarum strains* to establish a correlation between the presence of a chromosomal rearrangement and the ability to grow in high concentrations of sulfite. Sulfite tolerance was tested in plates containing different concentrations of potassium metabisulphite (MBS) ranging from 0 to 0.4 g/l, to compare with typical sulfite concentrations in wines (0.1–0.2 g/l) (S11 Fig). The results showed a significantly (*t*-test; p<0.001) higher MBS resistance of the strains with the VII$^{XVI}$ rearrangement in comparison with the strains with the ancestral type *SSU1* promoter (Figs 1D and S11). The strain with the XI$^{XVI}$ translocation also shows higher value than the stains without translocations. The resistance phenotype observed for the *S. uvarum* strains is similar to the resistance of the *S. cerevisiae* strains, showing significantly higher resistance, in the case of the strains with the chromosomal translocation, to similar sulfite levels [30,38]. Only the strains carrying any of the two reported chromosomal translocation events were able to grow in plates with the maximum concentration of MBS tested, while the maximum tolerable concentration of MBS of strains without the translocations was 0.2 g/l. This phenotypic characterization of the *S. uvarum* strains, together with the PCR amplification, allowed us to identify a clear correlation between the presence of a rearrangement in the *SSU1* promoter and the tolerance to sulfite (Fig 1D and S2 Table).

## Structural variations in the *SSU1* promoter are responsible for the overexpression of this gene

To confirm that the chromosomal translocation events in the *SSU1* promoter were leading to an increase of the expression of this gene, qPCR studies were performed with the *S. uvarum* strains. Fermentations with and without MBS were conducted with strains carrying the most frequent translocation (VII$^{XVI}$). We compared the *SSU1* expression of the wine BMV58 and CECT12600 strains against the *SSU1* expression of two strains with no chromosomal translocations: the strain CBS2986 [40], isolated from wine fermentation, and the natural NPCC1290 strain isolated from an *Araucaria araucana* tree [21]. Relative expression of the *SSU1* gene to the strain NPCC1314 (*SSU1* promoter without chromosomal translocations) was calculated

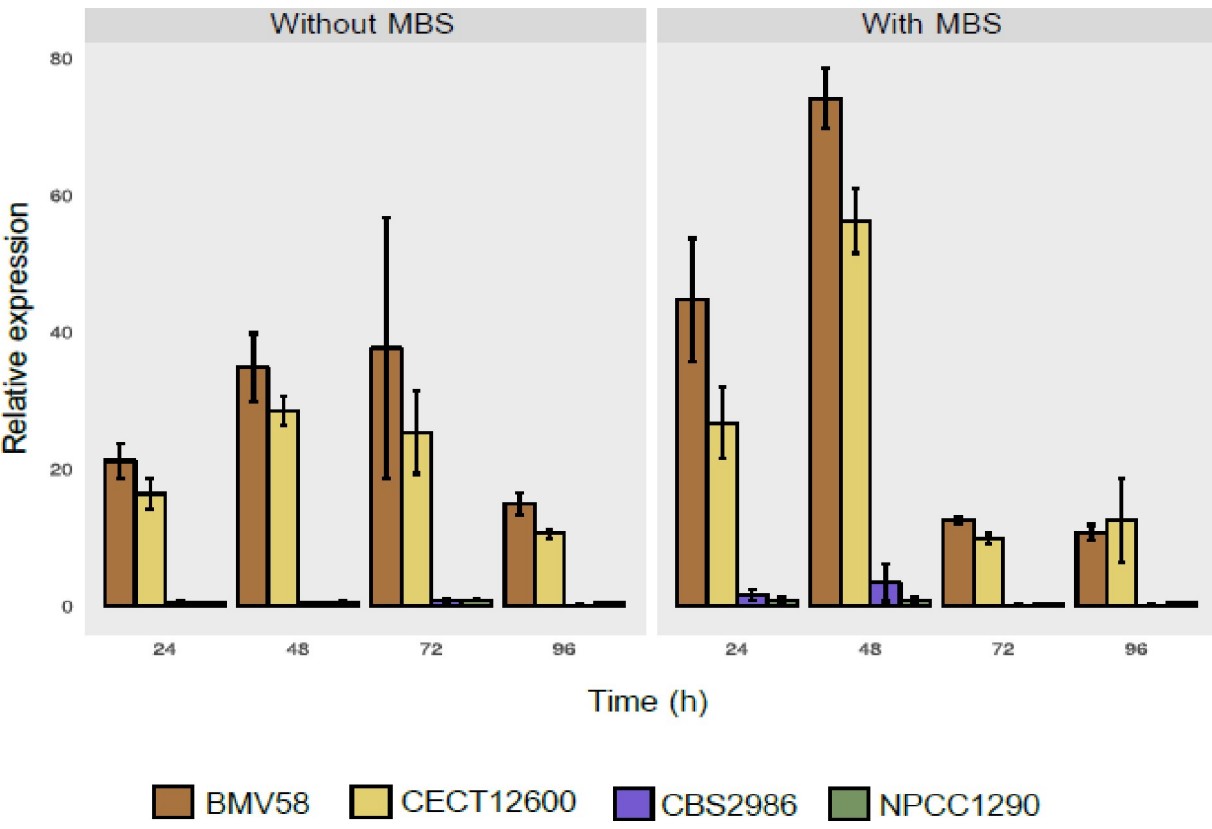

**Fig 2. Relative *SSU1* expression and growth in *S. uvarum* strains during fermentation.** Expression of the *SSU1* gene was studied during wine fermentation in synthetic must with or without sub lethal sulfite (MBS) concentration (15 mg/l) for two strains with the VII[XVI] translocation (BMV58 and CECT12600) and two with the ancestral chromosomes (NPCC1290 and CBS2986). Daily samples were taken until day four and, after mRNA extraction, *SSU1* gene expression was quantified by qPCR. Two constitutive genes (*ACT1* and *RDN18*) were used to normalize qPCR data. All expression measures were relativized to the *SSU1* expression in the NPCC1314 strain (ancestral *SSU1* promoter) grown under the same fermentation conditions.

(Fig 2 and S5 Table). The experiment was conducted with a low concentration of MBS (15 mg/l) to allow yeast to growth in contrast to the sulfite tolerance tests performed at higher concentrations (0 to 0.4 g/l). In this experiment, we observed a clear over-expression of the *SSU1* gene in the two strains with the translocation VII[XVI] when compared to the wild strain (NPCC1290) but also to the wine strain (CBS2986). This suggests that the chromosomal translocation at the *SSU1* promoter is a specific adaptation to sulfite presence rather than an adaptation to the wine environment. We also observed that the over-expression of the *SSU1* gene is not dependent on the presence of sulfite in the media. We performed a two-way analysis of variance (ANOVA) and both BMV58 and CECT12600 strains showed significantly higher expression levels than the other strains in the two conditions analyzed (with and without MBS), although expression was higher with MBS for all the strains, especially during the first two days of fermentation (Fig 2).

A second fermentation experiment was conducted to measure the *SSU1* expression of both BMV58 (VII[XVI]) and BR6-2 (XI[XVI]). Besides, to demonstrate the effect of the two different chromosomal translocation events in the *SSU1* gene expression, we obtained two modified versions of the *S. uvarum* type strain CBS7001, where the wild type *SSU1* promoter was substituted with the BMV58 or BR6-2 *SSU1* promoters. *SSU1* gene expression was also measured in these mutants together with the wild type CBS7001 (Fig 3).

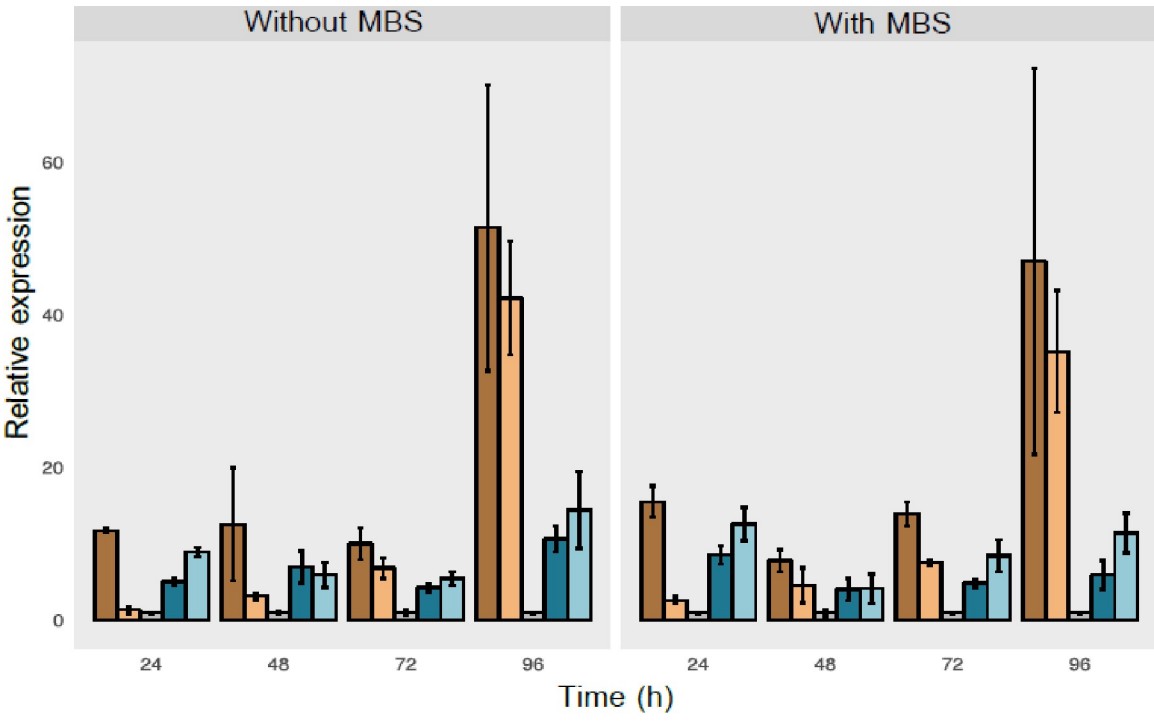

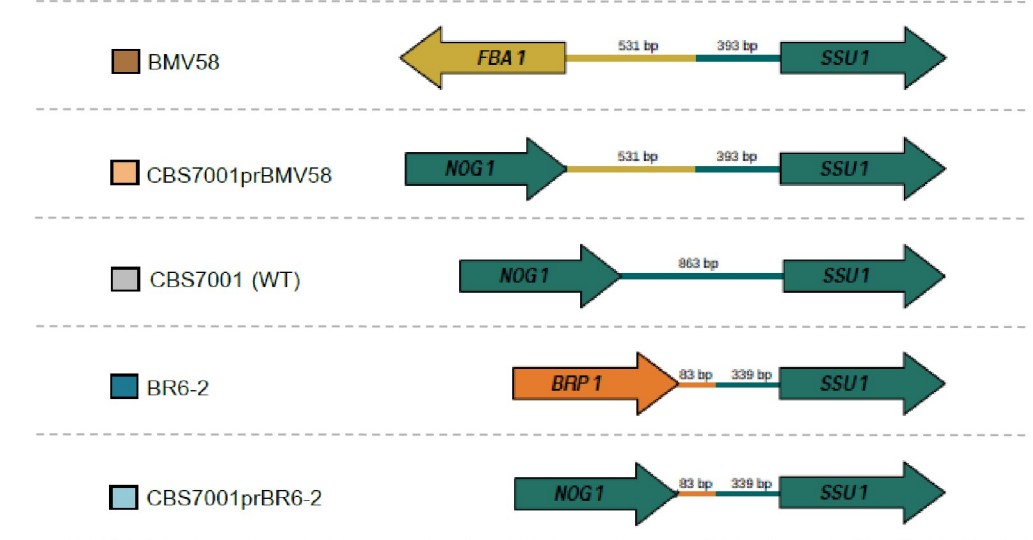

**Fig 3. Relative *SSU1* expression in *S. uvarum* wild type and edited strains grown in a fermentation experiment.** Expression of *SSU1* gene was studied during wine fermentation in synthetic must with or without sub lethal sulfite (MBS) concentration (15 mg/l) for a strain with the wild type chromosomes (CBS7001), a strain with the VII$^{XVI}$ translocation (BMV58), a strain with the VII$^{XVI}$ translocation (BR6-2), a modified version of the CBS7001 type strain with the BM58 *SSU1* promoter (CBS7001pBM58) and a modified version of the CBS7001 type strain with the BR6-2 *SSU1* promoter (CBS7001pBR6-2). A schematic representation of the different *SSU1* promoters is presented for each strain. Daily samples were taken until day four and, after mRNA extraction, *SSU1* gene expression was quantified by qPCR. Two constitutive genes (*ACT1* and *RDN18*) were used to normalize qPCR data. All expression measures were relativized to the *SSU1* expression in the CBS7011 wild type strain grown under the same fermentation conditions.

First, we confirmed that both types of chromosomal translocations generated *SSU1* overexpression compared to the wild-type strain (CBS7001). We observed that the BR6-2 *SSU1* promoter (in the CBS7001(prBR6-2) strain) produced an over-expression of *SSU1* not significantly different (*t*-test; p<0.05) than that observed for the strain BR6-2, except for time point 96 h with MBS. In the other case, the promoter of BMV58 (strain CBS7001(prBMV58)) generates a clearer over-expression in the *SSU1* levels compared with the CBS7001 strain, specially after the first 24 hours of fermentations. The overexpression of *SSU1* in the edited strain CBS7001(prBMV58) showed no significantly different values (*t*-test; p<0.05) compared to the strain BMV58 except at the 24 h time point without MBS and at 24 h and 72 h time points with MBS when the transcriptions levels were significantly lower when compared to the BMV58 strain. Although is clear that the new promoter of CBS7001(prBMV58) strain produces a significant overexpression of *SSU1* compared to the CBS7001 strain, these latter results suggest that other factors as the chromosomal context or other unknown upstream/downstream elements, not transferred to CBS7001(prBMV58) could have further influenced *SSU1* expression in the BMV58 strain. A similar trend was observed when we tested sulfite resistance of the recombinant strains (S10 Fig) since CBS7001(prBR6-2) reached a similar resistance than BR6-2 whereas CBS7001(prBMV58) showed higher resistance than the CBS7001 strain but not that much of BMV58 strain. It has to be noted that there is an appreciable difference in the expression of BMV58 between Figs 2 and 3 that correspond to a certain degree of variability observed in the *SSU1* expression data combined with the different normalization of the data but, in fact, unnormalized data showed no statistical differences between both datasets except time point 24 h in the experiment without MBS.

## Phylogenetic reconstruction and the origin of the *SSU1*-promoter chromosomal translocation events

A total number of 11 strains were found to have the chromosomal translocations described above. These strains were all isolated from wine or cider fermentations (S2 Table), anthropic environments where sulfite is commonly used as an antimicrobial preservative. Two of these strains were also isolated from Argentinean cider fermentation (as the strain NPCC1417). No chromosomal translocation events were found in the South American strains isolated from natural environments, neither in the ones isolated from *chicha*, a beverage performed in traditional fermentation with no sulfite addition.

To unravel the origin of the new chromosomal translocations discovered in this study we performed a phylogenetic analysis using whole-genome sequencing data from 21 strains. The selected strains represent different origins, populations, and *SSU1* promoter versions (ancestral, VII$^{XVI}$, or XI$^{XVI}$) (Fig 4). The phylogeny revealed that strains carrying chromosomal translocations in the *SSU1* promoter are located at different branches in the tree and they did not constitute a monophyletic group (Fig 4). It also revealed that the strains with translocations were not located at branches belonging to *S. uvarum* strains from Australasia or South America B populations, previously described by Almeida et al. [22].

South American and European strains appear as intermixed, including those South America A and Holarctic strains described by Almeida et al. [22]. Most of these branches showed low support values, indicating that other relationships are possible.

To further investigate the origin of the chromosomal rearrangement shared between the Argentinean NPCC1417 strain and the European wine strains, we estimated pairwise nucleotide divergences for the genes surrounding the *SSU1* promoter between BMV58 and NPCC1417, which share translocation, and between BMV58 and NPCC1309 and between BMV58 and NPCC1313, two other Argentinian strains without translocation, isolated in the

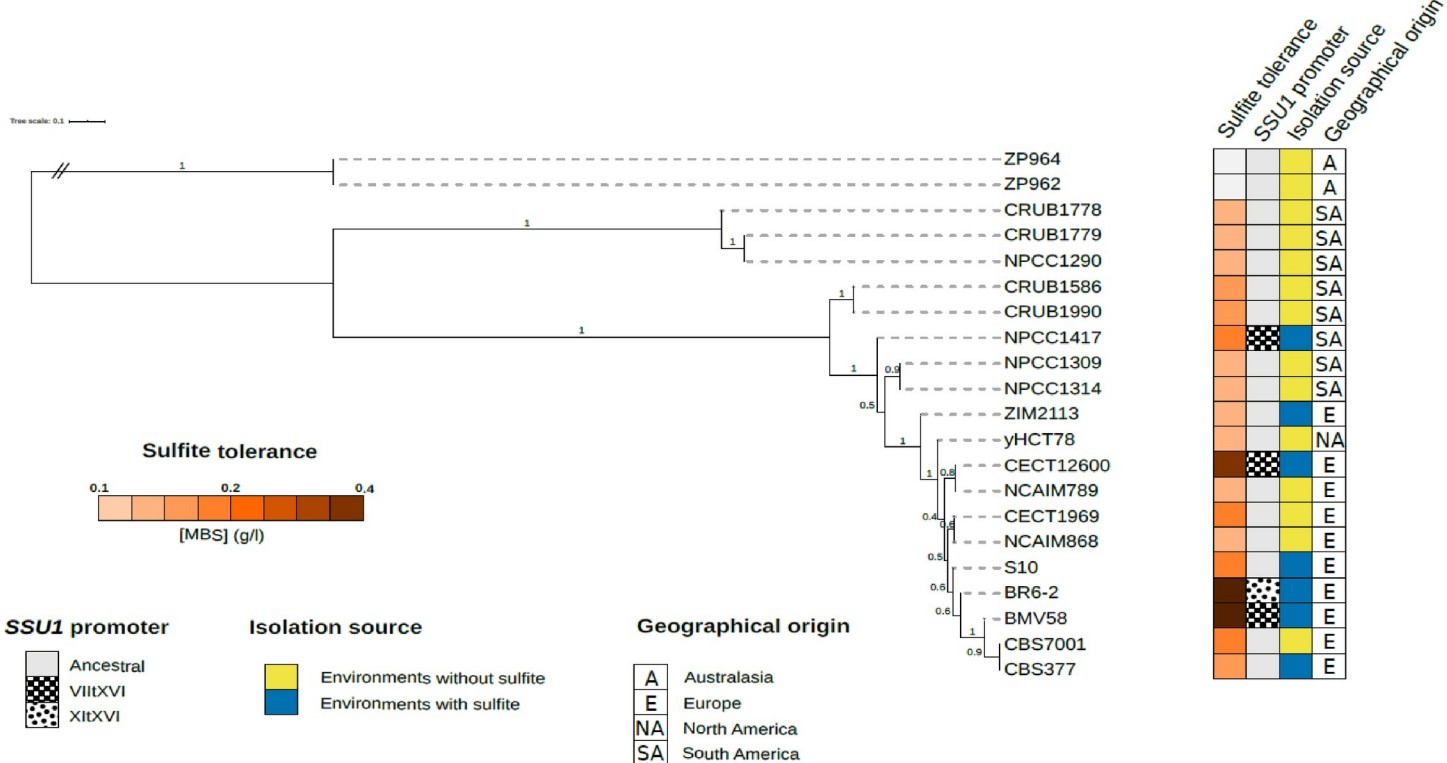

**Fig 4. Phylogenetic analysis of the *S. uvarum* sequenced genomes.** Phylogeny obtained with ASTRAL-III for 1265 unrooted individual gene trees shared among the 21 *S. uvarum* strains. Branch-support values, ranging from 0 to 1, are located at the nodes and represent the support for a quadripartition. The tree was rooted using ZP962 and ZP964 from the Australasian population as outgroups. A heatmap next to the branch labels shows the sulfite tolerance of the strains, *SSU1* promoter variant, and the isolation source. Sulfite tolerance was measured by drop test assay and it is color-coded from minimum (0 g/l) to maximum (0.4 g/l) MBS concentration. Strains were divided according to their isolation source taking into account whether the isolation environment contained sulfite used as a preservative or not. Finally, ancestral *SSU1* promoter strains (strains without any translocation in the promoter) and the two different chromosomal translocation events found are shown.

same geographic location as the NPCC1417 and they were closer in the phylogenetic tree. This analysis (S2 Fig) revealed that NPCC1417 and BMV58 share an identical segment of 117 kb, including 21 genes from the *SSU1* gene towards the right side (until YPL068C gene) and 33 genes from the *SSU1* gene towards the left side (until the YGL044C gene), which is divergent in the other Argentinian strains without translocation. In the reciprocally translocated chromosome (S3 Fig) the results showed a clearly shorter region of low genetic divergence around the breakpoint, which supports the action of selection on the new *SSU1* allele. The presence of this conserved segment of linkage disequilibrium is indicative that the translocations present in NPCC1417 and BMV58 share a common origin. To study how frequent was in the NPCC1417 genome to find genomic regions containing consecutive conserved genes with the BMV58, we randomly selected 1,000 windows of twenty genes along the genome of the NPCC1417 and calculated genetic distances against the BMV58 orthologous genes. A window of 20 genes containing all its pairwise distances equal to zero resulted significantly different from the distribution created from the 1,000 randomly selected windows ($p$-value $< 0.05$; Whitney-Wilcoxon test).

## Discussion

In this work, we present a case of a convergent adaptation of *S. uvarum* strains, isolated from fermentation environments, to grow in sulfite containing media, a preservative usually added

in industrial processes such as wine or cider fermentation. This is the first example reported in which different chromosomal rearrangements originated by two different chromosomal translocation events resulted in the over-expression of the *SSU1* gene and, therefore, an increase of the sulfite tolerance in the strains carrying the translocations.

In *S. cerevisiae*, different cases of structural variations have been described in the promoter of the *SSU1* gene. These variations include chromosomal [30,34,37], which involve different chromosomes than those reported for *S. uvarum*, and a chromosomal inversion [38]. These *SSU1* promoter variants described for *S. cerevisiae* have been reported to cause the overexpression of this gene being those strains much more tolerant to the presence of sulfites in the culture media. This is the first time that a chromosomal translocation event in the *SSU1* promoter, providing an adaptive value, is described for another *Saccharomyces* species, different from *S. cerevisiae*.

As far as we know, our work describes the first example of a phenotypic convergence produced by independent chromosomal rearrangements in two of the most divergent *Saccharomyces* species, *S. cerevisiae*, and *S. uvarum* (20% of nucleotide divergence). In fact, the last common ancestor existed 20 million years ago [12]. Strains of both species exhibit rearrangements at different locations in the promoter of the *SSU1* gene that allows adaptation to tolerate high sulfite concentrations. It is well known the enormous adaptive role that exerts the overexpression of the *SSU1* gene in industrial strains [35,36]. This effect would explain why it has been favored the appearance of molecular mechanisms, as the chromosomal translocation at the *SSU1* locus, resulting in a phenotypic convergence. Interestingly, the four chromosomal translocation events described so far are independent, produced at different locations of the *SSU1* promoter, and involving reciprocal translocations between chromosome XVI and different partners. Our results, including several complementary approaches, confirm the strong selection pressure that the antimicrobial effect of sulfite imposes on yeasts in human-driven fermentations, as well as remarks on the role of chromosomal rearrangements as a source of variation to promote yeast adaptations in fast-evolving environments.

The molecular mechanisms that produced the overexpression of the *SSU1* gene remains unclear. The regulation mechanism of the *SSU1* gene known until now is mediated by the five-zinc-finger transcription factor codified by the *FZF1* gene. This gene acts as a positive regulator of the *SSU1* by binding directly to its upstream promoter [26]. The Fzf1p binding sequence has been described as 5'-CTATCA-3'. This sequence is present at many sites throughout the genome but *SSU1* is the only demonstrated target. We have identified the binding sequence in the ancestral promoter *SSU1* version of strains without chromosomal rearrangements. Interestingly, both rearrangements described in this work, occurred before the *FZF1* binding site, like in *S. cerevisiae*, hence, the *SSU1* promoter region lost the Fzf1p binding site due to the chromosomal rearrangements. Our main hypothesis is that *FZF1* is not regulating the expression of the *SSU1* gene in these *S. uvarum* strains. Instead of that, this gene could be possibly constitutively active or being regulated by another of several transcription factors that have not been identified yet. We can also conclude from our experiments that the overexpression effect of the *SSU1* gene is not dependent on the presence of sulfite in the media as this gene is highly expressed from the early stages of fermentation with and without sulfite.

The XI^XVI translocation was found in a unique European strain isolated from a cider fermentation while the VII^XVI translocation event is shared among European and South American strains. Previous population analyses performed on the *S. uvarum* species classify them into four differentiated populations: Australasian, South America B, South America A, and Holarctic [22]. In a recent study [41], the existence of South America A population, genetically differentiated from the Holarctic population has been questioned and the authors suggest that these strains are the result of the genetic admixture of Holarctic and South America B strains.

This fact, together with the high incongruence observed in our phylogenic reconstruction, leads us to think that they should not be properly considered as two different populations because they are, indeed, a mixed population. This idea is supported by the shared chromosomal rearrangement described in this study between strains isolated in Europe and Argentina. We hypothesize that these strains probably coexisted at the same location. This rearrangement was spread by sexual reproduction among different strains and it became fixed later in those strains grown in human-related environments where sulfite is used as a microbial preservative. Our data suggest that the VII[XVI] recombination had a unique and recent origin in a European strain, and then, it was inherited by these South American strains due to hybridizations between European and South American strains. This premise is supported by the conserved region observed in the *SSU1* surrounding gene sequences of NPCC1417 with respect to the translocated regions of the European strains. The conservation of this large segment could be due to a reduction of the recombination rate between the translocated and the standard chromosome alleles in the regions flanking the translocation point or to genetic hitchhiking in the surroundings of the translocated *SSU1* gene as the target of selection. However, the fact that the conserved region surrounding the reciprocal translocation site is significantly smaller does not support a lower recombination rate in the regions flanking the translocation points and, hence, is compatible with the presence of a large, linked region swept along with the selectively favored recombinant *SSU1* allele.

Finally, our discovery highlights the role of the *SSU1* gene promoter as a hotspot of evolution at different taxonomic levels. *S. cerevisiae* is the predominant species in sulfite-containing environments as wine, cider, and other fermented beverages. However, *S. uvarum* can be also dominant in certain types of fermentation, especially those performed at lower temperatures [19,20,42]. This abundance can explain the detection of the *SSU1* locus chromosomal translocation events exactly in those species, as an adaptation to sulfite. Other species such as *Hanseniospora uvarum*, *Metschnikowia pulcherrima*, *Bretanomyces sp*. among others can be found in relatively high numbers in those environments at the beginning and even at more advanced stages of fermentations [43,44]. Future studies should examine chromosomal rearrangements involving the gene responsible for sulfite detoxification in these species.

## Materials and methods

### Yeast strains, media, and fermentations

Information about the yeast strains used in this study is summarized in S2 Table. Strains were maintained and propagated in GPYD media (5 g/L yeast extract, 5 g/L peptone, 20 g/L glucose). Wine fermentations were carried out in 100 mL bottles filled with 90 ml of synthetic must (100 g/L glucose, 100 g/L fructose, 6 g/L citric acid, 6 g/L malic acid, mineral salts, vitamins, anaerobic growth factors, 300 mg/L assimilable nitrogen) that simulates standard grape juice [45]. Fermentations were inoculated at $5.0 \times 10^6$ cells/ml density from overnight precultures determined by measuring $OD_{600}$. Bottles were closed with Muller valve caps and incubated at 25°C with gentle agitation. Fermentation progress was followed by daily measuring bottle weight loss. In the fermentations with MBS, after preliminary tests, a sub-lethal concentration (15 mg/l) of MBS that allow the four strains used (BMV58, CECT12600, NPCC1290, and NPCC1314) to grow was selected. All wine fermentations were performed at least in independent triplicates.

### Edited strains construction

To modify *SSU1* promoters in the CBS7001 strain we used the CRISPR-Cas9 technique as described by Generoso et al. [46]. Primers used are listed in S3 Table. The plasmid pRCCN

(Addgene) was used to target the *SSU1* promoter to integrate the recombinant fragments, amplified from BMV58 or BR6-2 strains. The protospacer sequences were chosen according to Doench et al. [47] using CBS7001 genome sequence as reference to avoid selecting unspecific gRNA. Then we amplified by PCR the plasmid pRRC-N, which carries the natMX resistance marker, with primers carrying the protospacer sequence at their 5' ends [46]. The PCR was carried out with Phusion High-Fidelity Polymerase following the provider instructions using the primers listed in S3 Table. Before addition to the transformation mix, we treated 30 μL of the PCR product with 10 U of DpnI restriction enzyme (Thermo Scientific) for 3 h to guarantee the degradation of pRRC-N template. To ensure the reparation by homologous recombination we used PCR amplified fragments of the *SSU1* promoter from BMV58 or BR6-2 strains whose 40 nucleotides of each side are homologous to both upstream and downstream sequences of the target sequence [48]. 1 mmol of the PCR fragment was added to the transformation mix, performed following Gietz and Schiestl method [49]. Transformants were selected in ClonNat (Sigma) GPY agar plates and verified by PCR using diagnostic primers (S3 Table) and sanger sequencing. Finally, the positive strains were cured of the pRCCN vector.

## Genome sequencing, assembly, and annotation

Strains were sequenced by Illumina HiSeq 2000 with paired-end reads of 100 bp long at the Genomics section from the Central Service of Experimental Research Support (SCSIE), University of Valencia. SPAdes [50], with default parameters, was used for *de novo* assembly.

BR6-2 strain and NPCC1314 were sequenced using PacBio sequencing Single Molecule, Real-Time (SMRT) DNA sequencing technology (platform: PacBio RS II; chemistry: P4-C2 for the pilot phase and P6-C4 for the main phase). The raw reads were processed using the standard SMRT analysis pipeline (v2.3.0). The *de novo* assembly was done using Flye (version 2.7) with 3 polishing iterations and default parameters [51].

MUMmer [52] was used to get the homology between the strains sequenced in this study and the reference *S. uvarum* strain CBS7001 [53]. This information was used to get scaffolds into chromosome structure (note that, in Scannel et al. [53] annotation, chromosome X was mislabeled as chromosome XII and vice-versa). Annotation was performed as described in [54]. We used a combination of two approaches including transferring the annotation from the *S. cerevisiae* S288c based on synteny conservation. The annotated assemblies were used to identify the ultrascaffolds containing the *SSU1* gene and the surrounding annotated genes. We identified the position of the *SSU1* gene and then we selected for further investigation those assemblies whose *SSU1* gene position and surrounding genes does not match with the reference strain position (chromosome XVI).

## Phylogenetic analyses

Annotated genomes sequenced in this study as well as collected data from previous studies [22,53] were used for phylogeny reconstruction. A list of the genomes used in this analysis can be found in supplementary S1 Table. Introgressed genes from other *Saccharomyces* species were removed from the analysis. A total number of 1265 orthologous genes were found among the 21 *S. uvarum* strains. Nucleotide sequences were translated into amino-acids and aligned with Mafft [55]. Aligned protein sequences were back-translated into codons. Maximum-Likelihood (ML) phylogeny reconstruction was performed for each gene using RAxML [56] with the GTRCAT model and 100 bootstrap replicates. ML-trees were concatenated to infer a coalescence-based phylogeny using ASTRAL-III, version 5.6.3 [57]. Tree was visualized using iTOL [58].

## Analyses of the origin of the shared chromosomal rearrangement among BMV58, CECT12600, and NPCC1417 strains

Gene sequences upstream and downstream of the *SSU1* gene were extracted to calculate genetic distances among the strains BMV58, CECT12600, and NPCC1417. Distances were calculated using the "dist.dna" function from the *ape* R package [59] under the "K81" model [60]. This method was repeated to calculate pairwise genetic distances using the BMV58 as a reference against NPCC1309 and NPCC1314 strains. An in-house python script was used to select 1,000 random windows of 20 genes within BMV58 and NPCC1417 genomes to calculated pairwise genetic distances.

## Southern blot analysis

We performed Southern blot analyses with karyotyping gels. Pulsed-field gel electrophoresis was performed under these conditions: 60 seconds during 12 h and 120 seconds during 14 h with an angle of 150˚ and a velocity of 6V/cm. The strains included were BMV58, CECT12600, NPCC1290, and NPCC1314. DNA was transferred to a nylon membrane Amersham Hybond -N+ (GE Healthcare Europe GmbH, Barcelona, Spain) according to manufactures protocol. We construct the probes using the primers listed in S3 Table and the PCR DIG Probe Synthesis Kit (Roche Applied Science, Mannheim, Germany). Each Southern blot analysis was done with high stringency conditions to be sure of the specificity of the probe. Hybridization was prepared with DIG Easy Hyb Granules (Roche Applied Science), following recommendations of the manufacturer for prehybridization, hybridization, and post hybridization washes. For washing, blocking, and detection of DIG-labeled probes DIG Wash and Block Buffer Set (Roche Applied Science) was used. For the detection of DIG-labeled molecules an Anti-Digoxigenin-AP, Fab fragment (1,10.000) (Roche Applied Science), was used. Finally, CDP-Star Set (Roche Applied Science), a chemiluminescent substrate for alkaline phosphatase was used at 1:100 dilution, and images were stored after 30 min of exposition.

## Gene expression determination

For each culture, a 10–20-ml sample was taken each day of wine fermentation. The cells were quickly collected by centrifugation, washed, and frozen with liquid $N_2$. Then, frozen cells were homogenized with a FastPrep-24 (MP Biomedicals, Santa Ana, USA) device with acid-washed glass beads (0.4 mm diameter; Sigma-Aldrich, Madrid, Spain) in LETS buffer (10 mm Tris pH 7.4, 10 mM lithium-EDTA, 100 mM lithium chloride, 1% lithium lauryl sulfate) for 30 s alternating with ice incubation six times. The phenol:chloroform method with minor modifications [61] was used to extract and purify total RNA. Then, cDNA was synthesized from the RNA and the expression of *SSU1* genes was quantified by qRT-PCR (quantitative real-time PCR). cDNA was synthesized in 13 μl using 2 μg of RNA mixed with 0.8 mM dNTP's and 80 pmol Oligo (dT). The mixture was incubated at 65˚C for 5 min and in ice for 1 min. Then, 5 mM dithiothreitol (DTT), 50 U of RNase inhibitor (Invitrogen, Waltham, USA), 1 × First-Strand Buffer (Invitrogen), and 200 U Superscript III (Invitrogen) were added in 20 μl mixture and this was incubated at 50˚C for 60 min and 15 min at 70˚C. qRT-PCR gene-specific primers (200 nM), designed (S3 Table) from consensus sequences between the different strains, were used in 10 μl reactions, using the Light Cycler FastStart DNA MasterPLUS SYBR green (Roche Applied Science) in a LightCycler 2.0 System (Roche Applied Science). All samples were processed for DNA concentration determination, amplification efficiency, and melting curve analysis. To obtain a standard curve, serial dilutions ($10^{-1}$ to $10^{-5}$) of a mixture of all samples was used. The average of *ACT1* and *RDN18-1* constitutive genes was used to normalize the amount of mRNA and to safeguard repeatability, correct interpretation, and accuracy [62].

## Sulfite tolerance assay

Sulfite tolerance was tested in YEPD +TA (tartaric acid) agar plates as described by Park et. al. [63]. YEPD (2% dextrose, 2% peptone and 1% yeast extract) was supplemented with L- tartaric acid at 75 mM buffered at pH 3.5 and potassium metabisulfite ($K_2S_2O_5$, MBS) was added to each plate to a final concentration of 0, 0.05, 0.10, 0.15, 0.20, 0.25, 0.30, 0.35 or 0.40, g/L. Yeast precultures were grown overnight in GPY medium. Cell cultures were diluted to $OD_{600}$ = 1. Then, serial 1:5 dilutions of cells were inoculated in MBS YEPD plates and incubated at 25˚C for a week.

## Supporting information

**S1 Table. Genome sequences used for the phylogenetic and variant calling analyses.** Collected data from previous works and de novo sequenced genomes used in this study.
(XLSX)

**S2 Table. Sulfite tolerance and *SSU1* promoter in a collection of *S. uvarum* strains isolated from different environments and geographic locations.** Drop test assay results are represented by the number of the most diluted (from 1 to the less diluted and 6 to the most diluted) that grew in each MBS concentration tested. The type of *SSU1* promoter is represented in the last column according to the results of PCR amplification.
(XLSX)

**S3 Table. List of primers used in this study**
(XLSX)

**S4 Table. Summary of the PCR tests to evaluate *SSU1* promoter configurations.**
(XLSX)

**S5 Table. Gene expression datasets including Ct values, ratios and normalizations.**
(XLSX)

**S1 Fig. Confirmation of the presence of the XVI^VII chromosomal translocation event in BMV58 and CECT12600 *S. uvarum* strains, comparing with the non-recombinant strains NPCC1290 and NPCC1314.** (A) A schematic representation of the chromosomal location of primers (arrows) and probes (purple rectangles) used to detect wild type (VII and XVI) and recombinant (VII^XVI and XVI^VII) chromosomes. Chromosomal size in Mbp is indicated in brackets. (B) PCR amplification used to test for the presence of wild type chromosomes VII (primers A-B) and XVI (primers C-D) or recombinant chromosomes VII^XVI (primers D-B) and XVI^VII (primers C-A). (C) Southern blots with chromosome VII and XVI left and right probes performed in genomic DNA obtained from BMV58, CECT12600, NPCC1290, and NPCC1314 *S. uvarum* strains. DNA fragment size is indicated in Mbp.
(PDF)

**S2 Fig. Determination of nucleotide divergences for the genes surrounding the *SSU1* promoter in the XVI^VII chromosome.** Pairwise genetic distances of the genes surrounding the *SSU1* promoter were calculated and represented in this study using BMV58 as reference. The x-axis represents the gene position using *SSU1* as reference (position 0). Green genes correspond to genes from the reference chromosome XVI and the orange gene corresponds to the reference chromosome VII.
(PDF)

**S3 Fig. Determination of nucleotide divergences for the genes surrounding the *SSU1* promoter in the VII^XVI chromosome.** Pairwise genetic distances of the genes surrounding the

*SSU1* promoter were calculated and represented in this study using BMV58 as reference. The x-axis represents the gene position using *SSU1* as reference (position 0). Green genes correspond to genes from the reference chromosome XVI and the orange gene corresponds to the reference chromosome VII.
(PDF)

**S4 Fig. PCR amplification gels for the different strains and the indicated primer combinations to test *SSU1* promoter translocations.**
(PDF)

**S5 Fig. PCR amplification gels for the different strains and the indicated primer combinations to test *SSU1* promoter translocations.**
(PDF)

**S6 Fig. PCR amplification gels for the different strains and the indicated primer combinations to test *SSU1* promoter translocations.**
(PDF)

**S7 Fig. PCR amplification gels for the different strains and the indicated primer combinations to test *SSU1* promoter translocations.**
(PDF)

**S8 Fig. PCR amplification gels for the different strains and the indicated primer combinations to test *SSU1* promoter translocations.**
(PDF)

**S9 Fig. PCR amplification gels for the different strains and the indicated primer combinations to test *SSU1* promoter translocations.**
(PDF)

**S10 Fig. Sulfite resistance test for the CRISPR edited strains CBS7001(prBMV58) and CBS7001(prBR6-2) in comparison to control strains CBS7001, T73 and BMV58.** Two MBS concentrations (0.1 and 0.2 g/l) were evaluated.
(PDF)

**S11 Fig. Sulfite resistance test for the strains used in this work.** Nine MBS concentrations (0.0–0.4 g/l) were evaluated.
(PDF)

## Acknowledgments

Genome sequences were obtained at the Genomics section from the Central Service of Experimental Research Support (SCSIE), University of Valencia. We also thank Chris Todd Hittinger, Philippe Marullo and Diego Libkind for kindly providing strains.

## Author Contributions

**Conceptualization:** Amparo Querol, Eladio Barrio, Christian Ariel Lopes, Roberto Pérez-Torrado.

**Funding acquisition:** Amparo Querol, Eladio Barrio, Christian Ariel Lopes.

**Investigation:** Laura G. Macías, Melisa González Flores, Ana Cristina Adam.

**Methodology:** Laura G. Macías, Melisa González Flores, Ana Cristina Adam, María E. Rodríguez.

**Software:** Laura G. Macías.

**Supervision:** María E. Rodríguez, Amparo Querol, Eladio Barrio, Christian Ariel Lopes, Roberto Pérez-Torrado.

**Writing – original draft:** Laura G. Macías, Roberto Pérez-Torrado.

**Writing – review & editing:** Laura G. Macías, Amparo Querol, Eladio Barrio, Christian Ariel Lopes, Roberto Pérez-Torrado.

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
