## [Decision Letter · Decision Letter 0]

8 Mar 2021

Dear Dr Pérez-Torrado,

Thank you very much for submitting your Research Article entitled 'Convergent adaptation of Saccharomyces uvarum to sulfite, an antimicrobial preservative widely used in human-driven fermentations' to PLOS Genetics.

The manuscript was fully evaluated at the editorial level and by independent peer reviewers. The reviewers appreciated the attention to an important problem, but raised some substantial concerns about the current manuscript. Based on the reviews, we will not be able to accept this version of the manuscript, but we would be willing to review a much-revised version. We cannot, of course, promise publication at that time.

If you decide to revise the manuscript for further consideration at PLOS Genetics, please aim to resubmit within the next 60 days, unless it will take extra time to address the concerns of the reviewers, in which case we would appreciate an expected resubmission date by email to plosgenetics@plos.org.

[LINK]

We are sorry that we cannot be more positive about your manuscript at this stage. Please do not hesitate to contact us if you have any concerns or questions.

Yours sincerely,

Justin C. Fay

Associate Editor

PLOS Genetics

Kirsten Bomblies

Section Editor: Evolution

PLOS Genetics

The reviewers appreciated your findings and thought the conclusions were well supported. However, they also identified a number of issues that should be addressed. After reading the reviews I believe they have all given fair and thoughtful comments that can be addressed and will improve your work. I also have a few specific editorial comments.

1) One of the reviewers would like to see the work better placed into the larger picture of convergent evolution. I agree and believe it would be worth providing some context on how often convergence occurs through translocations and on a short time-scale. Convergence (or rather parallel evolution) in experimental evolution doesn't seem that relevant. But you may be able to argue your case is interesting because it involves translocations (which are quite rare) and occurred during a very short time interval since sulfites have been used.

2) All three reviewers found the section on the selective sweep unclear in regards to the idea and how it is supported. After rereading this section I believe the intent is to show that the region around SSU1 VIItXVI has very little variation among the three strains having the translation, showing that the translocation is very young. A genome or chromosome plot of pairwise divergence would better convey this point. As it stands, the description of what was done/found is vague, e.g. "having more genes with nucleotide distances equal to zero". This could be 2 vs 1 gene which is not convincing. Also, why focus on coding sequences, which are often conserved, rather than all sequences extending from the breakpoint.

3) The reviewers also noted multiple areas where clarity and improvement in presentation could be made. And, they noted a number of points where there is missing information needed to replicating the study. In revising your work, please ensure that these relevant details are included. Also, please check the English for errors.

Reviewer's Responses to Questions

**Comments to the Authors:**

Reviewer #1: Macias et al, present a manuscript that describes the evolution of sulfite resistance by convergent mechanisms in S. uvarum. This evolution occurs by chromosome translocation that modifies the SSU1 promoter and promotes increased transcription level of the gene. Overall, there is not much to say about this paper as the results are fairly straightforward. Whether the results were unexpected is a matter of opinion - similar more extreme cases have been documented by Ken Wolfe’s lab on mating-type switching or by Sandy Johnson’s lab on mating-type specific expression though the exact mechanism of convergence may be different. Other simple aneuploidies have been observed throughout the tree of life as measures of adaptation.

Though most results support the conclusion of the paper, I find the presentation to be confusing. There are many sections in the result sections that don’t appear to have anything to do with the paper, and many sentences are stated without a clear interpretation of what the analysis means. I list here suggestions of locations where the text can be severely improved.

1) The section on phylogenetic reconstruction includes a rather large description of the phylogenetic tree. Every strain on the tree appears in the text, with a mention of their isolation source, their country of origin, and some hyperboles on the positioning of these strains. What is the meaning of all of this? Why can’t the authors just put the country/fermentation in the figure? It doesn’t seem like the large descriptive text from lines 220 to 240 to be useful.

2) Line 257. I don’t understand what was done here. What is the SSU1 promoter of surrounding genes?

3) Line 262. A word appears to be missing here.

4) At the end of line 267, a concluding sentence is necessary here.

5) I’m not familiar with how selective sweeps due to chromosomal rearrangements can occur. The sweeps the authors are talking about are due to sexual recombination, and I understand fairly well how a beneficial mutation can result in a sweep and loss of genetic diversity around the mutation. However, chromosomal rearrangements can often result in strange issues, such as mating isolation and non-viability. What is the proposed mechanism here for the spread of a translocation through sexual reproduction? Are there no essential genes in these translocated regions?

Is it possible that recombination is blocked at regions of chromosomal translocations, which would negate the evidence of a sweep in the traditional sense? Regions of inversions frequently block recombination, and it wouldn’t be surprising to me if it also occurred close to translocations.

6) What is GPY medium?

7) Figure 4: the 3 shades of grey for the types of chromosomal rearrangements should be changed. There’s no gradient needed for this (there is no difference in resistance for the two anyway).

8) Is it strange that SSU1 is not induced in the presence of MBS in WT strains but only induced in the strains with a chromosomal translocation? The authors have a large section on Fzf1 being the inducer of SSU1 but there is no evidence that this gene is even expressed in the WT strain (as shown in Fig 2 and 3).

9) Figure 3 can be made clearer with a diagram of the promoter conformations that is being reconstructed.

10) There is a typo on figure 4 (“without”)

11) Can the authors standardize their concentration units? Both molarity and mg/L are used for MBS. What is the typical concentration of sulfite in wine? Are the concentrations/resistance here relevant? How does the resistance compare to S. cerevisiae used in wine cultures?

12) Line 192 to 210 is a word by word reconstruction of the graph we see in Figure 3. Can this be compressed somehow?

Overall, some clarifications to the manuscript are required. Possible discussions of the caveats on the interpretation of results (#5), and further manuscript compression would make the paper clearer. Though I don’t disagree that parallel evolution to SSU1 overexpression has happened in S. uvarum, I think it is a bit of an exaggeration to say that the results were unexpected and I fail to appreciate how the findings here are placed in context with other modes of convergent evolution that have been previously observed in these fungal lineages.

Reviewer #2: Macias et al. demonstrate an example of convergent evolution in adaptation to wine fermentation. Previous work has shown that wine strains of Saccharomyces cerevisiae have reciprocal translocation events that result in a chimeric promoter of the gene SSU1 and increase sulfite resistance. Here, the authors demonstrate a similar phenomenon in the distantly related species Saccharomyces uvarum, which is primarily used in wine and cider fermentation at low temperatures. They find two different translocation events in wine and cider strains of S. uvarum that also result in a recombinant SSU1 promoter, present in 11 different strains of this species. They demonstrate that strains with translocations have an increased tolerance to sulfite, and show that strains with the translocation have increased constitutive and inducible gene expression of SSU1. I think this manuscript highlights an interesting result of convergent evolution involving several independent translocations in S. cerevisiae and S. uvarum, but several conclusions could benefit from additional experiments, analysis, and explanation, elaborated below.

Major Comments

I had several questions related to the expression data. First, can the authors address the time sensitivity of the expression of SSU1? Why might it be elevated or decreased at different time points? What might be responsible for the drastic difference in expression of BMV58 between Figure 2 and 3? Second, the promoter expression experiments have a lot of variability between the promoter construct strain and the strain that it’s meant to mimic, particularly for the BMV58 strain. I think this data does support the conclusion that the translocation upstream of SSU1 influences expression, but the authors should further address potential discrepancies driving these differences (e.g., strain background differences, potentially not including all important regions in the promoter construct, etc.). Perhaps illustrating with a figure like Figure 1C what the promoter construct includes. I would suggest to further clarify this, the authors could test sulfite tolerance in the CBS7001 wild type and the 2 promoter constructs.

I’m having a hard time understanding the hypothesis of the selective sweep in relation to a translocation. The authors mention that the SSU1 translocation may be responsible for reproductive isolation between strains without the translocation, however, the strains with the translocation are not monophyletic. In Figure S2, the panel is showing a region that spans the translocation, but only one of the strains shown has the translocation, so I don’t understand what is being compared (e.g., for the strains without the translocation, is this comparing the syntenic regions on VII and XVI separately?). Inclusion of more strains in this analysis is needed (minimally all the other strains that have a translocation), and an extension beyond the described region, and including the other breakpoint for the translocation. More support and/or explanation is needed to strengthen this conclusion.

Minor comments

Introduction to convergent evolution - the examples provided from Arabidopsis and Drosophila are not what I would typically understand as convergent evolution. Can the authors elaborate more on their definition and how these examples illustrate the concept?

Throughout the manuscript – I feel like the word “translocation” would be a better description of the events than “recombinations” (or alternatively chimeric promoter or recombinant promoter)

Line 77- sensibility should be “sensitivity”

Lines 155-157: Authors state that sulfite tolerance is significantly different between strains with the translocation and strains without, but no statistics are reported. Furthermore, the authors report that strains with the translocation can grow at 8mM sulfite concentration, but Figure 1D only shows up to 4 mM. Perhaps a different figure type would be more appropriate for representation of this data, particularly because from this figure it appears that BR62- has the strongest tolerance to sulfite, but from my understanding, some of the strains with the other rearrangement have a higher tolerance.

Lines 175-176 – “carrying out the recombination VIIXVI” should be “with the translocation VIIXVI”

193 – Delete, “Second,”

206-210 – should clarify what they mean when they say unless the new promoter produces significant overexpression. Why would the chromosomal context or other elements not be important in the case of overexpression? And by overexpression, are they referring to specific time points like 96 hours?

270 – “the case of a convergent adaptation of S. uvarum strains” to “a case of convergent adaptation of S. uvarum strains”

287 – nucleotide divergence is ~20%

289-291 – I’m not sure what the authors are trying to say with this sentence

313 – “depending” should be “dependent”

350-353 – rephrase to “Future studies should examine chromosomal rearrangements involving the gene responsible for sulfite detoxification in these species.”

Figure 4 – typo “Environments withouth sulfite”

Reviewer #3: Macias et al report an interesting case of convergent adaptation to sulfite in various strains of S. uvarum. Resistance to sulfites induced by structural chromosomal changes leading to overexpression of SSU1 has been extensively studied in S. cerevisiae, but this manuscript represents the first description of this phenomenon in another species of Saccharomyces.

One major concern is that most of the raw results are not made available which makes difficult to assess the robustness of the analyses:

- The authors should provide the PCR amplification gels on the 61 strains as a supplementary figure.

- The authors should also check by southern blot whether the rearrangement between chromosome XI and XVI corresponds to a reciprocal translocation event.

- Similarly, the drop test assays on MBS should be presented.

- The CT values of the qPCR experiments should be made available.

- How the substitutions of the WT SSU1 promoter by the BMV58 and BR6-2 SSU1 promoters were achieved in the reference background should be described with more details and controls.

In addition, it would be worth mentioning in the introduction that nearly 600 S.cerevisiae strains from fermented grape juice were genotyped for the presence of the reciprocal translocation between chromosomes VIII and XVI (Marullo et al. Front Microbiol, 2020). Another interesting finding that could be worth mentioning is that in the absence of a pre-existing triplication in the ECM34 promoter, this translocation was shown to promote sulfite sensitivity in the BY laboratory strain background (Fleiss et al. Plos Genet 2018).

There are other questions, listed below, that would deserve to be answered before this manuscript is suitable for publication.

- In SupTable 1, the number of annotated genes ranges from 1369 to 5666. Where does such a large discrepancy come from?

- In addition, the sequencing and assembly statistics for the 21 genomes should be provided (coverage, N50, number of contigs, nb of scaffolds, % of the reference genome covered, etc).

- One general question about the 21 sequenced strains would be to know their ploidy and their heterozygosity levels because diploid genomes could be heterozygous for the translocations. Given that a single allele can only be represented in the genome assemblies, it is possible that other cases of translocation remain undetected.

- Line 115: The authors should better explain how they identified the rearrangements “Assemblies allowed us to identify two candidate chromosomal rearrangements in the promoter of this gene located at chromosome XVI”

- Line 129 : change VIItXVI by VIIItXVI

- In Fig 1C, the interesting information would not so much be the identity of the gene upstream SSU1 after the translocation but rather the identity of the new promoter brought in front of SSU1 by the rearrangement. Are they any genome-wide expression data available that would allow to look for the expression pattern of these promoters in a WT configuration?

- Line 256: The authors propose that a selective sweep of 117 kb occurred surrounding the SSU1 promoter region between the NPCC1417 and BMV58 strains. I don’t think that the term selective sweep is well chosen because it is hard to imagine that selection was able to eliminate all the genetic diversity across such a large chromosomal segment. The conservation between the chromosome configuration and the DNA sequence between the 2 strains most likely result from a recent admixture between the 2 backgrounds as suggested in the discussion. In addition, the author should also provide a hypothesis on the evolutionary origin of the translocation in the CECT12600 strain.

Finally, the word ‘recombination’ is not correctly used throughout the text and should be replaced by ‘rearrangement’ in many instances. In general, I think the English could be improved.

**Have all data underlying the figures and results presented in the manuscript been provided?**

Reviewer #1: Yes

Reviewer #2: Yes

Reviewer #3: **No: **see details in the 'response to authors'

PLOS authors have the option to publish the peer review history of their article (what does this mean?). If published, this will include your full peer review and any attached files.

Reviewer #1: No

Reviewer #2: No

Reviewer #3: No

---

## [Decision Letter · Decision Letter 1]

31 Aug 2021

Dear Dr Pérez-Torrado,

Thank you very much for submitting your Research Article entitled 'Convergent adaptation of Saccharomyces uvarum to sulfite, an antimicrobial preservative widely used in human-driven fermentations' to PLOS Genetics.

The manuscript was fully evaluated at the editorial level and by independent peer reviewers. The reviewers appreciated the attention to an important topic but identified some concerns that we ask you address in a revised manuscript

We therefore ask you to modify the manuscript according to the review recommendations. Your revisions should address the specific points made by each reviewer.

[LINK]

Yours sincerely,

Justin C. Fay

Associate Editor

PLOS Genetics

Kirsten Bomblies

Section Editor: Evolution

PLOS Genetics

Reviewer's Responses to Questions

**Comments to the Authors:**

Reviewer #1: The revised manuscript has addressed all my previous concerns.

I have a few comments for clarity:

1) Figure 3: The orders of the bars on the histogram should be the same order as the legend below.

2) Figure S2 is missing from this submission

Reviewer #2: I appreciate that the authors added several new components to their manuscript to address reviewer concerns (including PacBio sequencing to identify reciprocal translocation, sulfite tolerance of CBS7001 and promoter constructs, etc), and I like the new addition to Figure 3.

The idea that recombination is reduced around the translocation breakpoints is logical to me, and I don’t want to belabor this, but I think the analysis presented and the response to reviewers on this topic could still use further work. At the least, the same analysis should be conducted at the other breakpoint of the translocation for the given strains. If my understanding is correct, we would predict that the other breakpoint would show a similar signature. This section could use more support from the literature about what is known/expected about patterns of recombination and diversity around breakpoints. Ideally other strains with the same translocation would be compared to identify if the same blocks are conserved. I do not see how the presented data could be consistent with hitchhiking, so if the authors think this is plausible, they need to elaborate on this point.

A few other minor points:

I have a better understanding of the presented expression data now. I appreciate the authors answering my queries regarding the time course and variation in expression, as well as the differences between Fig. 2 and 3. I would suggest that the authors incorporate what is known about the time course and expression into the text. I have one additional comment here – the figure legends read 15 mg/L concentration but most other assays are done with a concentration 10x that or more, including the newly added Figure S4. From Figure S4, the promoter construct CBS7001(prBMV58) clearly cannot explain sulfite resistance in higher levels of BMS. I see the authors note this in the results, might also be worth mentioning more clearly that expression was not tested at high concentrations.

A couple of the supplementary files mentioned in the text seem to be missing from this version (Figures S2, S5)

I would suggest the authors include in the materials and methods or in their Table S1 the assembly statistics for the PacBio sequencing and if anything was done besides manual inspection to confirm the reciprocal translocation.

The paragraph on convergent evolution in the introduction is improved, but could still use a bit more tweaking. I have included it here edited for grammar, and suggest that the authors try to clarify more in their examples what they mean by convergent evolution. For example, in the insect example to toxic compounds, the authors could state what molecular mechanisms were found to underlie this adaptation, and thus how it was concluded that different lineages independently evolved this phenotype. In the yeast examples, the authors could specify with more details, such as point mutations in different genes resulted in the same phenotype - increased tolerance to X, etc.

Organisms belonging to different lineages can evolve independently to overcome similar

environmental pressures through different molecular mechanisms. This phenomenon, known as convergent evolution, is considered evidence of the action of natural selection [1,2]. In recent

years, comparative genomics studies have suggested that convergent adaptation occurs

more frequently than previously expected [3,4]. For example, species of insects spanning multiple

orders have independently evolved higher tolerance to toxic compounds

produced by plants [5], demonstrating that convergent adaptation can

occur in nature between organisms belonging to different taxonomic levels. In the case

of yeasts, convergent evolution by point mutations has been described both in evolving

yeast species in nature [6] and in short-term evolutionary studies in the species

Saccharomyces cerevisiae [7], for example in populations evolved under glucose

limitation [8]. Convergent evolution can occur through different mechanisms, including point

mutations, gene duplications, and interspecific hybridization. Examples of convergent evolution via chromosomal rearrangements are rare, a single study has suggested that an intrachromosomal translocation is responsible of a convergent evolution in independent lineages in the case of the major histocompatibility complex [9]. A second study has suggested that amylase evolution in fish may have converged though a putative chromosomal translocation, although this has not yet been confirmed [10].

Reviewer #3: The authors have responded satisfactorily to all my requests. I am wondering though whether the new PacBio data were made publicly available.

**Have all data underlying the figures and results presented in the manuscript been provided?**

Reviewer #1: Yes

Reviewer #2: Yes

Reviewer #3: Yes

PLOS authors have the option to publish the peer review history of their article (what does this mean?). If published, this will include your full peer review and any attached files.

Reviewer #1: No

Reviewer #2: No

Reviewer #3: No

---

## [Editor Report · Decision Letter 2]

11 Oct 2021

Dear Dr Pérez-Torrado,

We are pleased to inform you that your manuscript entitled "Convergent adaptation of Saccharomyces uvarum to sulfite, an antimicrobial preservative widely used in human-driven fermentations" has been editorially accepted for publication in PLOS Genetics. Congratulations!

Yours sincerely,

Justin C. Fay

Associate Editor

PLOS Genetics

Kirsten Bomblies

Section Editor: Evolution

PLOS Genetics

Comments from the reviewers (if applicable):

**Data Deposition**

http://datadryad.org/submit?journalID=pgenetics&manu=PGENETICS-D-21-00158R2

**Press Queries**

---

## [Editor Report · Acceptance letter]

4 Nov 2021

PGENETICS-D-21-00158R2 

Convergent adaptation of Saccharomyces uvarum to sulfite, an antimicrobial preservative widely used in human-driven fermentations 

Dear Dr Pérez-Torrado, 

We are pleased to inform you that your manuscript entitled "Convergent adaptation of Saccharomyces uvarum to sulfite, an antimicrobial preservative widely used in human-driven fermentations" has been formally accepted for publication in PLOS Genetics! Your manuscript is now with our production department and you will be notified of the publication date in due course.

With kind regards,

Katalin Szabo

PLOS Genetics

On behalf of:
